# With Friends Like These, Who Needs Adversaries?

**Saumya Jetley**[*1] **Nicholas A. Lord**[*1,2] **Philip H.S. Torr**[1,2]

[1]Department of Engineering Science, University of Oxford
[2]Oxford Research Group, FiveAI Ltd.
{sjetley, nicklord, phst}@robots.ox.ac.uk

## Abstract

The vulnerability of deep image classification networks to adversarial attack is now well known, but less well understood. Via a novel experimental analysis, we illustrate some facts about deep convolutional networks for image classification that shed new light on their behaviour and how it connects to the problem of adversaries. In short, the celebrated performance of these networks and their vulnerability to adversarial attack are simply two sides of the same coin: the input image-space directions along which the networks are most vulnerable to attack are the same directions which they use to achieve their classification performance in the first place. We develop this result in two main steps. The first uncovers the fact that classes tend to be associated with specific image-space directions. This is shown by an examination of the class-score outputs of nets as functions of 1D movements along these directions. This provides a novel perspective on the existence of universal adversarial perturbations. The second is a clear demonstration of the tight coupling between classification performance and vulnerability to adversarial attack within the spaces spanned by these directions. Thus, our analysis resolves the apparent contradiction between accuracy and vulnerability. It provides a new perspective on much of the prior art and reveals profound implications for efforts to construct neural nets that are both accurate and robust to adversarial attack.[1]

## 1 Introduction

Those studying deep networks find themselves forced to confront an apparent paradox. On the one hand, there is the demonstrated success of networks in learning class distinctions on training sets that seem to generalise well to unseen test data. On the other, there is the vulnerability of the very same networks to adversarial perturbations that produce dramatic changes in class predictions despite being counter-intuitive or even imperceptible to humans. A common understanding of the issue can be stated as follows: "While deep networks have proven their ability to distinguish between their target classes so as to generalise over unseen natural variations, they curiously possess an Achilles heel which must be defended." In fact, efforts to formulate attacks and counteracting defences of networks have led to a dedicated competition [1] and a body of literature already too vast to summarise in total.

In the current work we attempt to demystify this phenomenon at a fundamental level. We base our work on the geometric decision boundary analysis of [2], which we reinterpret and extend into a framework that we believe is simpler and more illuminating with regards to the aforementioned paradoxical behaviour of deep convolutional networks (DCNs) for image classification. Through a fairly straightforward set of experiments and explanations, we clarify what it is that adversarial examples represent, and indeed, what it is that modern DCNs do and do not currently do. In doing so, we tie together work which has focused on adversaries *per se* with other work which has sought to characterise the feature spaces learned by these networks.

---

[*]S. Jetley and N.A. Lord have contributed equally and assert joint first authorship.
[1]Source code for replicating all experiments is provided at https://github.com/torrvision/whoneedsadversaries.

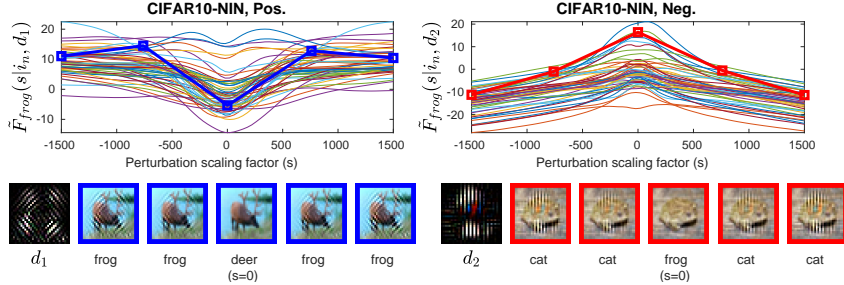

Figure 1: Plots of the 'frog' class score $\tilde{\mathcal{F}}_{frog}(s|\mathbf{i}_n, \mathbf{d}_j, \theta)$ for the Network-in-Network [3] architecture trained on CIFAR10, associated with two specific image-space directions $\mathbf{d}_1$ and $\mathbf{d}_2$ respectively. These directions are visualised as 2D images in the row below; the method of estimating them is explained in Sec. 3. Each plot corresponds to a randomly selected CIFAR10 test image $\mathbf{i}_n$. Adding or subtracting components along $\mathbf{d}_1$ causes the network to change its prediction to 'frog': as can be seen, a 'deer' with a mild diamond striping added to it gets classified as a 'frog'. This happens with little regard for the choice of input image $\mathbf{i}_n$ itself. Likewise, perturbations along $\mathbf{d}_2$ change *any* 'frog' to a 'non-frog' class: notice the predicted labels for the sample images along the red curve in the second plot. These class-transition phenomena are predicted by the framework developed in this paper. While simplistic functions along directions $\mathbf{d}_1$ and $\mathbf{d}_2$ are used by the network to accomplish the task of classification, perturbations along the very same directions constitute adversarial attacks.

Let $\check{\mathbf{i}}$ represent vectorised input images and $\bar{\mathbf{i}}$ be the average vector-image over a given dataset. Then, the mean-normalised version of the dataset is denoted by $\mathbb{I} = \{\mathbf{i}_1, \mathbf{i}_2, \cdots \mathbf{i}_N\}$, where the $n^{\text{th}}$ image $\mathbf{i}_n = \check{\mathbf{i}}_n - \bar{\mathbf{i}}$. We define the perturbation of the image $\mathbf{i}_n$ in the direction $\mathbf{d}_j$ as: $\tilde{\mathbf{i}}_n \leftarrow \mathbf{i}_n + s\hat{\mathbf{d}}_j$, where $s$ is the perturbation scaling factor and $\hat{\mathbf{d}}_j$ is the unit-norm vector in the direction $\mathbf{d}_j$. The image is fed through a network parameterised by $\theta$ and the output score[2] for a specific class $c$ is given by $\mathcal{F}_c(\tilde{\mathbf{i}}|\theta)$. This class-score function can be rewritten as $\mathcal{F}_c(\mathbf{i}_n + s\hat{\mathbf{d}}_j|\theta)$, which we equivalently denote by $\tilde{\mathcal{F}}_c(s|\mathbf{i}_n, \mathbf{d}_j, \theta)$. Our work examines the nature of $\tilde{\mathcal{F}}_c$ as a function of movement $s$ in specific image-space directions $\mathbf{d}_j$ starting from randomly sampled natural images $\mathbf{i}_n$, for a variety of classification DCNs. With this novel analysis, we uncover three noteworthy observations about these functions that relate directly to the phenomenon of adversarial vulnerability in these nets, all of which are on display in Fig. 1. We now discuss these observations in more detail.

Before we begin, we note that these directions $\mathbf{d}_j$ are obtained via the method explained in Sec. 3 and by design exhibit either positive or negative association with a specific class. In Fig. 1 we study two such directions for the 'frog' class: similar directions exist for all other classes. Firstly, notice that the score of the corresponding class $c$ ('frog', in this case) as a function of $s$ is often approximately symmetrical about some point $s_0$, *i.e.* $\tilde{\mathcal{F}}_c(s - s_0|\mathbf{i}_n, \mathbf{d}_j, \theta) \approx \tilde{\mathcal{F}}_c(-s - s_0|\mathbf{i}_n, \mathbf{d}_j, \theta) \, \forall s$, and monotonic in both half-lines. This means that simply increasing the magnitude of correlation between the input image and a single direction causes the net to believe that more (or less) of the class $c$ is present. In other words, the image-space direction sends all images either towards or away from the class $c$. In the former scenario, the direction represents a class-specific universal adversarial perturbation (UAP). Second, let $i^{\mathbf{d}} = \mathbf{i} \cdot \hat{\mathbf{d}}$, and let $\mathbf{i}^{\mathbf{d}\perp}$ be the projection of $\mathbf{i}$ onto the space normal to $\hat{\mathbf{d}}$, such that $\mathbf{i}^{\mathbf{d}\perp} = \mathbf{i} - i^{\mathbf{d}}\hat{\mathbf{d}}$. Then, our results illustrate that there exists a basis of image space containing $\hat{\mathbf{d}}$ such that the class-score function is approximately additively separable *i.e.* $\mathcal{F}_c(\mathbf{i}|\theta) = \mathcal{F}_c([i^{\mathbf{d}}, \mathbf{i}^{\mathbf{d}\perp}]|\theta) \approx \mathcal{G}(i^{\mathbf{d}}) + \mathcal{H}(\mathbf{i}^{\mathbf{d}\perp})$ for some functions $\mathcal{G}$ and $\mathcal{H}$. This means that the directions under study can be used to alter the nets' predictions almost independently of each other. However, despite these facts, their 2D visualisation reveals low-level structures that are devoid of a clear semantic link to the associated classes, as shown in Fig. 1. Thus, we demonstrate that the learned functions encode a more simplistic notion of class identity than DCNs are commonly assumed to represent, albeit one that generalises to the test distribution to an extent. Unsurprisingly, this does not align with the way in which the human visual system makes use of these data dimensions: 'adversarial vulnerability' is simply the name given to this disparity and the phenomena derived from it, with the universal adversarial perturbations of [4] being a particularly direct example of this fact.

Finally, we show that nets' classification performance and adversarial vulnerability are inextricably linked by the way they make use of the above directions, on a variety of architectures. Consequently,

efforts to improve robustness by "suppressing" nets' responses to components in these directions (*e.g.* [5]) cannot simultaneously retain full classification accuracy. The features and functions thereof that DCNs currently rely on to solve the classification problem *are*, in a sense, their own worst adversaries.

## 2 Related Work

### 2.1 Fundamental developments in attack methods

Szegedy *et al.* coined the term 'adversarial example' in [7], demonstrating the use of box-constrained L-BFGS to estimate a minimal $\ell_2$-norm additive perturbation to an input image to cause its label to change to a target class while keeping the resulting image within intensity bounds. Strikingly, they locate a small-norm (imperceptible) perturbation at every point, for every network tested. Further, the adversaries thus generated are able to fool nets trained differently to one another, even when trained with different subsets of the data. Goodfellow *et al.* [8] subsequently proposed the 'fast gradient sign method' (FGSM) to demonstrate the effectiveness of the local linearity assumption in producing the same result, calculating the gradient of the cost function and perturbing with a fixed-size step in the direction of its sign (optimal under the linearity assumption and an $\ell_\infty$-norm constraint). The DeepFool method of Moosavi-Dezfooli *et al.* [9] retains the first-order framework of FGSM, but tailors itself precisely to the goal of finding the perturbation of *minimum* norm that changes the class label of a given natural image to any label other than its own. Through iterative attempts to cross the nearest (linear) decision boundary by a tiny margin, this method records successful perturbations with norms that are even smaller than those of [8]. In [4], Moosavi-Dezfooli & Fawzi *et al.* propose an iterative aggregation of DeepFool perturbations that produces "universal" adversarial perturbations: *single* images which function as adversaries over a large fraction of an *entire* dataset for a targeted net. While these perturbations are typically much larger than individual DeepFools, they do not correspond to human perception, and indicate that there are *fixed* image-space directions along which nets are vulnerable to deception *independently* of the image-space locations at which they are applied. They also demonstrate some generalisation over network architectures.

Sabour & Cao *et al.* [10] pose an interesting variant of the problem: instead of "label adversaries", they target "feature adversaries" which minimise the distance from a particular guide image in a selected network feature space, subject to a constraint on the $\ell_\infty$-norm of image-space distance from a source image. Despite this constraint, the adversarial image mimics the guide very closely: not only is it nearly always assigned to the guide's class, but it appears to be an inlier with respect to the guide-class distribution in the chosen feature space. Finally, while "adversaries" are conceived of as small perturbations applied to natural images such that the resulting images are still recognisable to humans, the "fooling images" of Nguyen *et al.* [11] are completely unrecognisable to humans and yet confidently predicted by deep networks to be of particular classes. Such images are easily obtained by both evolutionary algorithms and gradient ascent, under direct encoding of pixel intensities (appearing to consist mostly of noise) and under CPPN [12]-regularised encoding (appearing as abstract mid-level patterns).

### 2.2 Analysis of adversarial vulnerability and proposed defences

In [13], Wang *et al.* propose a nomenclature and theoretical framework with which to discuss the problem of adversarial vulnerability in the abstract, agnostic of any actual net or attack thereof. They denote an oracle relative to whose judgement robustness and accuracy must be assessed, and illustrate that a classifier can only be both accurate and robust (invulnerable to attack) relative to its oracle if it learns to use exactly the same feature space that the oracle does. Otherwise, a network is vulnerable to adversarial attack in precisely the directions in which its feature space departs from that of the oracle. Under the assumption that a net's feature space contains some spurious directions, Gao *et al.* [5] propose a subtractive scheme of suppressing the neuronal activations (*i.e.* feature responses) which change significantly between the natural and adversarial inputs. Notably, the increase in robustness is accompanied by a loss of performance accuracy. An alternative to network feature suppression is the compression of input image data explored in *e.g.* [14, 15, 16].

Goodfellow *et al.* [8] hypothesise that the high dimensionality and excessive linearity of deep networks explain their vulnerability. Tanay and Griffin [17] begin by taking issue with the above via illustrative toy problems. They then advance an explanation based on the angle of intersection

of the separating boundary with the data manifold which rests on overfitting and calls for effective regularisation - which they note is neither solved nor known to be solvable for deep nets. A variety of training-based [8, 18, 19, 20] methods are proposed to address the premise of the preceding analyses. Hardening methods [8, 18] investigate the use of adversarial examples to train more robust deep networks. Detection-based methods [19, 20] view adversaries as outliers to the training data distribution and train detectors to identify them as such in the intermediate feature spaces of nets. Notably, these methods [19, 20] have not been evaluated on the feature adversaries of Sabour & Cao *et al.* [10]. Further, data augmentation schemes such as that of Zhang et al. [21], wherein convex combinations of input images are mapped to convex combinations of their labels, attempt to enable the nets to learn smoother decision boundaries. While their approach [21] offers improved resistance to single-step gradient sign attacks, it is no more robust to iterative attacks of the same type.

Over the course of the line of work in [2], [22], [23], and [24], the authors build up an image-space analysis of the geometry of deep networks' decision boundaries, and its connection with adversarial vulnerability. In [23], they note that the DeepFool perturbations of [9] tend to evince relatively high components in the subspace spanned by the directions in which the decision boundary has a high curvature. Also, the sign of the mean curvature of the decision boundary in the vicinity of a DeepFooled image is typically reversed with respect to that of the corresponding natural image, which provides a simple scheme to identify and undo the attack. They conclude that a majority of image-space directions correspond to near-flatness of the decision boundary and are insensitive to attack, but along the remaining directions, those of significant curvature, the network is indeed vulnerable. Further, the directions in question are observed to be *shared* over sample images. They illustrate in [2] why a hypothetical network which possessed this property would theoretically be *predicted* to be vulnerable to universal adversaries, and note that the analysis suggests a direct construction method for such adversaries as an alternative to the original randomised iterative approach of [4]: they can be constructed as random vectors in the subspace of shared high-curvature dimensions.

## 3  Method

The analysis begins as in [2], with the extraction of the principal directions and principal curvatures of the classifier's image-space class decision boundaries. Put simply, a principal direction vector and its associated principal curvature tell you how much a surface curves as you move along it in a particular direction, from a particular point. Now, it takes many decision boundaries to characterise the classification behaviour of a multiclass net: $\binom{C}{2}$ for a $C$-class classifier. However, in order to understand the boundary properties that are useful for discriminating a given class from all others, it should suffice to analyse only the $C$ 1-vs.-all decision boundaries. Thus, for each class $c$, the method proceeds by locating samples very near to the decision boundary $(\mathcal{F}_c - \mathcal{F}_{\hat{c}}) = 0$ between $c$ and the union of remaining classes $\hat{c} \neq c$. In practice, for each sample, this corresponds to the decision boundary between $c$ and the closest neighbouring class $\tilde{c}$, which is arrived at by perturbing the sample from the latter ("source") to the former ("target"). Then, the geometry of the decision boundary is estimated as outlined in Alg. 1 below[3], closely following the approach of [2]:

---
**Algorithm 1** Computes mean principal directions and principal curvatures for a net's image-space decision surface.

---
**Input:** network class score function $\mathcal{F}$, dataset $\mathbb{I} = \{\mathbf{i}_1, \mathbf{i}_2, \cdots \mathbf{i}_N\}$, target class label $c$
**Output:** principal curvature basis matrix $\mathbf{V}_b$ and corresponding principal curvature-score vector $\mathbf{v}_s$
    **procedure** PRINCIPALCURVATURES($\mathcal{F}, \mathbb{I}, c$)
        $\overline{\mathbf{H}} \leftarrow$ null
        **for** each sample $\mathbf{i}_n \in \mathbb{I}$ s.t. $\arg\max_k(\mathcal{F}_k(\mathbf{i}_n)) \neq c$ **do**
            $\hat{c} \leftarrow \arg\max_k(\mathcal{F}_k(\mathbf{i}_n))$             ▷ network predicts $\mathbf{i}_n$ to be of class $\hat{c}$
            $\mathcal{H}_{c\hat{c}}$: define as Hessian of function $(\mathcal{F}_c - \mathcal{F}_{\hat{c}})$       ▷ subscripts select class scores
            $\tilde{\mathbf{i}}_n \leftarrow$ DEEPFOOL($\mathbf{i}_n, c$)           ▷ approximate nearest boundary point to $\mathbf{i}_n$
            $\overline{\mathbf{H}} \leftarrow \overline{\mathbf{H}} + \mathcal{H}_{c\hat{c}}(\tilde{\mathbf{i}}_n)$         ▷ accumulate Hessian at sample boundary point
        $\overline{\mathbf{H}} \leftarrow \overline{\mathbf{H}}/\|\mathbb{I}\|$          ▷ normalise mean Hessian by number of samples
        $(\mathbf{V}_b, \mathbf{v}_s) =$ EIGS($\overline{\mathbf{H}}$)        ▷ compute eigenvectors and eigenvalues of mean Hessian
        **return** $(\mathbf{V}_b, \mathbf{v}_s)$

---

The authors of [2] advance a hypothesis connecting positively curved directions with the universal adversarial perturbations of [4]. Essentially, they demonstrate that if the normal section of a net's decision surface along a given direction can be locally bounded on the outside by a circular arc of

a particular positive curvature in the vicinity of a sample image point, then geometry accordingly dictates an upper bound on the distance between that point and the boundary in that direction. If such directions and bounds turn out to be largely common across sample image points (which they do), then the existence of universal adversaries follows directly, with higher curvature implying lower-norm adversaries. This argument is depicted visually in the supplementary material. It is from this point that we move beyond the prior art and begin an iterative loop of further analysis, experimentation, and demonstration, as follows.

## 4 Experiments and Analysis

Provided only that the second-order boundary approximation holds up well over a sufficiently wide perturbation range and variety of images, the model implies that the distance of such adversaries from the decision boundary should increase as a function of their norm. Also, the attack along any positively curved direction should in that case be associated with the corresponding target class: the class $c$ in the call to Alg. 1. And while positively curved directions may be of primary interest in [2], the extension of the above geometric argument to the negative-curvature case points to an important corollary: as sufficient steps along positive-curvature directions should perturb increasingly into class $c$, so should steps along *negative*-curvature directions perturb increasingly *away* from class $c$. Finally, the plethora of approximately zero-curvature (flat) directions identified in [23, 2] should have negligible effect on class identity.

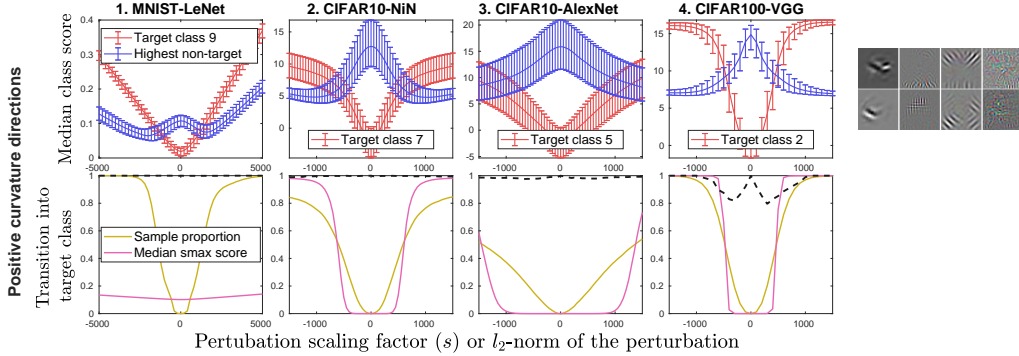

Figure 2: Selected class scores plotted as functions of the scaling factor $s$ of the perturbation along the most positively curved direction per net. The 'Median class score' plot compares the score of a randomly selected target class with the supremum of the scores for the non-target classes. Each curve represents the median of the class scores over the associated dataset, bracketed below by the 30th-percentile score and above by the 70th. The 'Transition into target class' plot depicts the fraction of the dataset *not* originally of the target class, but which is transitioned into the target class by the perturbation. Alongside, we graph that population's median softmax target-class score. The black dashed line represents the fraction of the population originally of the target class that remains in the target class under the perturbation. The image grid on the right illustrates the 2D visualisations of the two most-positively curved directions for randomly selected target classes: the columns correspond, from left to right, with the four net-dataset pairs under study. To observe class scores as functions of the norms of the perturbations along the most negatively curved and flat directions, refer to the supplement.

### 4.1 Class identity as a function of the component in specific image-space directions

To test how well the above conjectures hold in practice, we graph statistics of the target and non-target class scores over the dataset as a function of the magnitude of the perturbation applied in directions identified as above. The results are depicted in Fig. 2, in which the predicted phenomena are readily evident. Along the selected positive-curvature directions, as the perturbation magnitude increases (with either sign), the population's target class score approaches and then surpasses the highest non-target class score. The monotonicity of this effect is laid bare by graphing the fraction of non-target samples perturbed into the target class, alongside the median target class softmax score. Note, again, that the link between the directions in question and the target class identity is established *a priori* by Alg. 1. We continue in the supplementary material and show that, as predicted, the same phenomenon is evident in reverse when using negative-curvature directions instead. All that changes is that it is the population's non-target class scores that overtake its target class score with increasing

perturbation norm, with natural samples of the target class accordingly being perturbed out of it. We also illustrate the point that flatness of the decision boundary manifests as flatness of both target and non-target class scores: over a wide range of magnitudes, these directions do not influence the network in any way. While Fig. 2 illustrates these effects at the level of the population, Fig. 1 shows a disaggregation into individual sample images, with one response curve per sample from a large set. The population-level trends remain evident, but another fact becomes apparent: empirically, the shapes of the curves change very little between most samples. They shift vertically to reflect the class score contribution of the orthonormal components, but they themselves do not otherwise much depend on those components. That is to say that at least some key components are approximately additively separable from one another. This fact connects directly to the fact that such directions are "shared" across samples in the first place, and thus identifiable by Alg. 1.

A more intuitive picture of what the networks are actually doing begins to emerge: they are identifying the high-curvature image-space directions as features associated with respective class identities, with the curvature magnitude representing the sensitivity of class identity to the presence of that feature. But if this is true, it suggests that what we have thus identified are actually the directions which the net relies on *generally* in predicting the classes of natural images, with the curvatures-cum-sensitivities representing their relative weightings. Accordingly, it should be possible to disregard the "flat" directions of near-zero curvature without any noticeable change in the network's class predictions.

### 4.2 Network classification performance versus effective data dimensionality

To confirm the above hypothesis regarding the relative importance of different image-space directions for classification, we plot the training and test accuracies of a sample of nets as a function of the subspace onto which their input images are projected. The input subspace is parametrised by a dimensionality parameter $d$, which controls the number of basis vectors selected per class. We use four variants of selection: the $d$ most positively curved directions per class (yielding the subspace $S_{pos}$); the $d$ most negatively curved directions per class (yielding the subspace $S_{neg}$); the union of the previous two (subspace $S_{neg \cup pos}$); and the $d$ least curved (flattest) directions per class (subspace $S_{flat}$). The subspace $S$ so obtained is represented by the orthonormalised basis matrix $\mathbf{Q}_d$ (obtained by QR decomposition of the aggregated directions), and each input image $\mathbf{i}$ is then projected[4] onto $S$ as $\mathbf{i}^d = \mathbf{Q}_d \mathbf{Q}_d^\top \mathbf{i}$. Accuracies on $\{\mathbf{i}^d\}$ as a function of $d$ are shown in the top row of Fig. 3.

The outcome is striking: it is evident that in many cases, classification decisions have effectively already been made based on a relatively small number of features, corresponding to the most curved directions. The sensitivity of the nets along these directions, then, is clearly learned purposefully from the training data, and does largely generalise in testing, as seen. Note also that at this level of analysis, it essentially does not matter whether positively or negatively curved directions are chosen. Another important point emerges here. Since it is the high-curvature directions that are largely responsible for determining the nets' classification decisions, the nets should be vulnerable to adversarial attack along *precisely these directions*.

### 4.3 Link between classification and adversarial directions

It has already been noted in [23] that adversarial attack vectors evince high components in subspaces spanned by high-curvature directions. We expand the analysis by repeating the procedure of Sec. 4.2 for various attack methods, to determine whether existing attacks are indeed exploiting the directions in accordance with the classifier's reliance on them. Results are displayed in the bottom row of Fig. 3, and should be compared against the row above. The graphs in these figures illustrate the direct relationship between the fraction of adversarial norm in given subspaces and the corresponding usefulness of those subspaces for classification. The inclusion of the saliency images of [25] alongside the attack methods makes explicit the fact that adversaries are themselves an exposure of the net's notion of saliency.

By now, two results hint at a simpler and more direct way of identifying bases of classification/adversarial directions. First, a close inspection of the class-score curves sampled and displayed in Fig. 1 reveals a direct connection between the curvature of a direction near the origin and its derivative magnitude over a fairly large interval around it. Second, this observation is made more

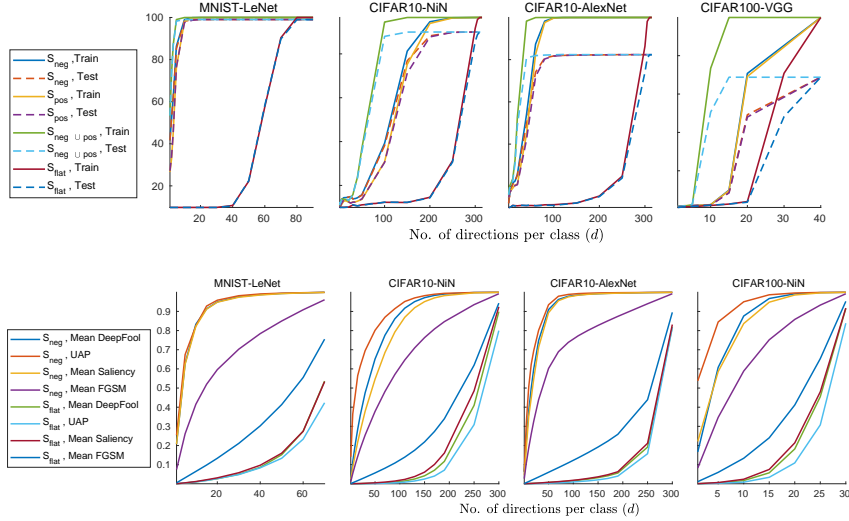

Figure 3: *Top row:* Training and test classification accuracies for various DCNs on image sets projected onto the subspaces described in Sec. 4.2, as a function of their dimensionality parameter $d$ (from 0 until the input space is fully spanned). The principal directions defining the subspaces are obtained by applying Alg. 1 once for each possible choice of target class $c$ and retaining $d$ directions per class. Note the relationship between the ordering of curvature magnitudes and classification accuracy by comparing the $S_{flat}$ curves to the others. *Bottom row:* Mean $\ell_2$-norms of various adversarial perturbations (DeepFool [9], FGSM [8] and UAP [4]) and saliency maps [25] when projected onto the same subspaces as above, as a fraction of their original norms.

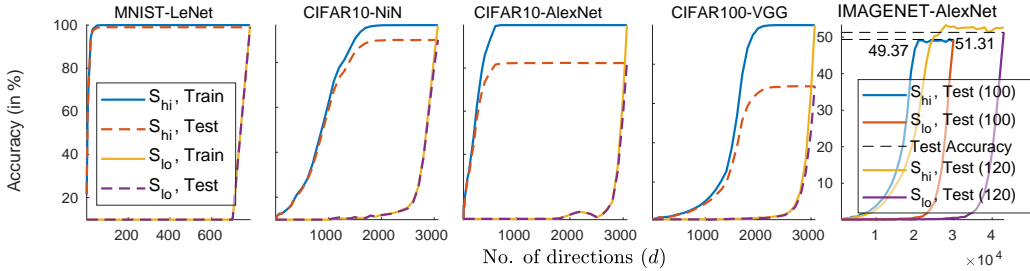

Figure 4: Classification accuracies on image sets projected onto subspaces of the spans of their corresponding DeepFool perturbations. For each net-dataset pair, DeepFool perturbations are computed over the image set and assembled into a matrix that is decomposed into its SVD. The singular vectors are ordered as per their singular values: $S_{hi}$ represents the high-to-low ordering, $S_{lo}$ the low-to-high, and $d$ the number of vectors retained. Compare this figure to Fig. 3 (while noticing how $d$ now counts the total number of directions). For the ImageNet experiments, owing to memory constraints, the SVD is performed on downsampled DeepFools of size $100 \times 100 \times 3$ and $120 \times 120 \times 3$, respectively. The resulting singular vectors span the entire effective classification space of correspondingly downsampled images. This is evinced by the fact that the classification accuracy of images projected onto the singular vectors' subspace saturates to the same performance as that yielded when the net is tested directly on the downsampled images.

clear in Fig. 3 where it can be seen that the directions obtained by boundary curvature analysis in Alg. 1 correspond to the directions exploited by various *first*-order methods. Thus, we hypothesise that to identify such a basis, one need actually only perform SVD on a matrix of stacked class-score gradients[5]. Here, we implement this using a collection of DeepFool perturbations to provide the required gradient information, and repeat the analysis of Sec. 4.2, using singular values to order the vectors. The results, in Fig. 4, neatly replicate the previously seen classification accuracy trends for high-to-low and low-to-high curvature traversal of image-space directions. Henceforth, we use these directions directly, simplifying analysis and allowing us to analyse ImageNet networks.

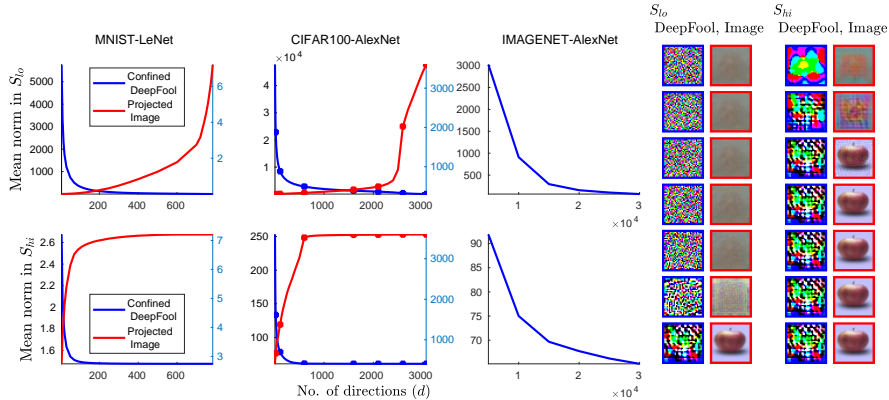

Figure 5: Blue curves depict the mean $\ell_2$-norms of "confined DeepFool" perturbations: those that are calculated under strict confinement to the respective subspaces of Fig. 4, also detailed in Sec. 4.3. Note the differences in scale of the $y$-axes of the different plots. For MNIST and CIFAR, we also plot (in red) the mean norms of the projections of the input images onto those subspaces: observe the inverse relationship between the two curves. The columns on the right visualise, from top to bottom, sample images at the indicated points on the curves in the CIFAR100-AlexNet plots, from left to right: blue-bordered images represent confined DeepFool perturbations (rescaled for display), with their red-bordered counterparts displaying the projection of the corresponding sample CIFAR image onto the same subspace. Observe that when the human-recognisable object appearance is captured in any given subspace, the corresponding DeepFool perturbation becomes maximally effective (*i.e.* small-norm). Likewise, when the projected image is not readily recognisable to a human, the DeepFool perturbation is large. The feature space *per se* does not account for adversariality: the issue is in the net's response to the features.

While Fig. 3 displays the magnitudes of components of pre-computed adversarial perturbations in different subspaces, we also design a variation on the analysis to illustrate how effective an efficient attack method (DeepFool) is when *confined* to the respective subspaces. This is implemented by simply projecting the gradient vectors used in solving DeepFool's linearised problem onto each subspace before otherwise solving the problem as usual. The results, displayed in Fig. 5, thus represent DeepFool's "earnest" attempts to attack the network as efficiently as possible *within* each given subspace. It is evident that the attack *must* exploit genuine classification directions in order to achieve low norm.

| $d_{low}$ | $E_n\{\ell_2\ norm(\mathbf{i}_n)\}$ | $E_n\{\ell_2\ norm(\boldsymbol{\delta i}_n)\}$ | Accuracy (%) | Fooling rate (%) | | | | | |
|---|---|---|---|---|---|---|---|---|---|
| | | | | $f=1$ | $f=2$ | $f=3$ | $f=4$ | $f=5$ | $f=10$ |
| 227 | 26798.72 | 63.96 | 57.75 | 100.00 | 100.00 | 100.00 | 100.00 | 100.00 | 100.00 |
| 200 | 26515.20 | 53.19 | 55.80 | 32.75 | 77.25 | 88.95 | 92.20 | 94.35 | 97.65 |
| 150 | 26327.03 | 46.86 | 53.50 | 35.55 | 58.35 | 77.90 | 85.95 | 89.25 | 95.65 |
| 120 | 26159.98 | 41.92 | 51.75 | 36.15 | 49.80 | 66.20 | 76.90 | 82.95 | 92.90 |
| 100 | 26008.02 | 37.98 | 48.10 | 41.65 | 49.25 | 59.95 | 68.05 | 74.80 | 88.30 |

Table 1: The images $\mathbf{i}_n$ used to train AlexNet operate at the scale of $d_{orig} = 227$ (pixels on a side). In the pre-processing step, these images are downsized to $d_{low}$, before being upsampled back to the original scale. The reconstructed DeepFool perturbations $\boldsymbol{\delta i}_n$ lose some of their effectiveness, as seen in the fooling-rate column for $f = 1$. When the effect of downsampling is countered by increasing the value of the $\ell_2$-norms of these perturbations (using higher values of $f$), their efficacy is steadily restored. Note that the mean norms of images and perturbations are estimated in the upscaled space, as are the classification accuracies. The accuracy values for $d_{low} = \{100, 120\}$ should be compared to those at convergence in Fig. 4. Any difference in the performance scores is strictly due to the random selection of the subset of 2000 test images used for evaluation.

## 4.4 On image compression and robustness to adversarial attack

The above observations have made it clear that the most effective directions of adversarial attack are also the directions that contribute the most to the DCNs' classification performance. Hence, any attempt to mitigate adversarial vulnerability by discarding these directions, either by compressing the input data [14, 15, 16] or by suppressing specific components of image representations at intermediate network layers [5], must effect a loss in the classification accuracy. Further, our framework anticipates the fact that the nets must remain just as vulnerable to attack along the remaining directions that

continue to determine classification decisions, given that the corresponding class-score functions, which possess the properties discussed earlier, remain unchanged. We use image downsampling as an example data compression technique to illustrate this effect on ImageNet.

We proceed by inserting a pre-processing unit between the DCN and its input at test time. This unit downsamples the input image $\mathbf{i}_n$ to a lower size $d_{low}$ before upsampling it back to the original input size $d_{orig}$. The resizing (by bicubic interpolation) serves to reduce the effective dimensionality of the input data. For a randomly selected set of 2000 ImageNet [26] test images, we observe the change in classification accuracy over different values of $d_{low}$, shown in column 4 of Table 1. The fooling rates[6] for the downsampled versions of these natural images' adversarial counterparts, produced by applying DeepFool to the *original* network (without the resampling unit), follow in column 5 of the table. At first glance, it appears that the downsampling-based pre-processing unit has afforded an increase in the network robustness at a moderate cost in accuracy. Results pertaining to this tradeoff have been widely reported [14, 15, 5]. Here, we take the analysis a step further.

To start, we note the fact that the methodology just described represents a transfer attack from the original net to the net as modified by the inclusion of the resampling unit. As DeepFool perturbations $\delta\mathbf{i}_n$ are not designed to transfer in this manner, we first augment them by simply increasing their $\ell_2$-norm by a scalar factor $f$. We adjust $f$ from unity up to a point at which the mean DeepFool perturbation norm is still a couple of orders of magnitude smaller than the mean image norm, such that the perturbations remain largely imperceptible. The corresponding fooling rates grow steadily with respect to $f$, as is observable in Table 1. Hence, although the original full-resolution perturbations may be suboptimal attacks on the resampling variants of the network (as some components are effectively lost to projection onto the compressed space), sufficient rescaling restores their effectiveness. On the other hand, the modified net continues to be equally vulnerable along the remaining effective classification directions, and can easily be attacked *directly*. To go about this, we simply take the SVD of the stack of downsampled DeepFool perturbations, for $d_{low}$ values of 100 and 120 (owing to memory constraints). The resulting singular vectors span the entire space of classification/adversarial directions of the corresponding resampling network, as can be seen from the accuracy values in the rightmost subplot of Fig. 4. More crucially, lower-norm DeepFools can be obtained by restricting the attack's iterative linear optimisation procedure to the space spanned by these compressed perturbations, exactly as described in Sec. 4.3 and displayed in Fig. 5. This subspace-confined optimisation is analogous to designing a white-box DeepFool attack for the new network architecture inclusive of the resampling unit, instead of the original network, and is as effective as before. Note that this observation is consistent with the results reported in [16], where the strength of the examined gradient-based attack methods increases progressively as the targeted model better approximates the defending model.

## 5    Conclusion

In this work, we expose a collection of directions along which a given net's class-score output functions exhibit striking similarity across sample images. These functions are nonlinear, but are *de facto* of a relatively constrained form: roughly axis-symmetric[7] and typically monotonic over large ranges. We illustrate a close relationship between these directions and class identity: many such directions effectively encode the extent to which the net believes that a particular target class is or is not present. Thus, as it stands, the predictive power and adversarial vulnerability of the studied nets are intertwined owing to the fact that they base their classification decisions on rather simplistic responses to components of the input images in specific directions, *irrespective* of whether the source of those components is natural or adversarial. Clearly, any gain in robustness obtained by suppressing the net's response to these components must come at the cost of a corresponding loss of accuracy. We demonstrate this experimentally. We also note that these robustness gains may be lower than they appear, as the network actually remains vulnerable to a properly designed attack along the remaining directions it continues to use. A discussion including some nuanced observations and connections to existing work that follow from our study can be found in the supplementary material. To conclude, we believe that for any scheme to be truly effective against the problem of adversarial vulnerability, it must lead to a fundamentally more insightful (and likely complicated) use of features than presently occurs. Until then, those features will continue to be the nets' own worst adversaries.

**Acknowledgements.** This work was supported by the ERC grant ERC-2012-AdG 321162-HELIOS, EPSRC grant Seebibyte EP/M013774/1 and EPSRC/MURI grant EP/N019474/1. We would also like to acknowledge the Royal Academy of Engineering, FiveAI, and extend our thanks to Seyed-Mohsen Moosavi-Dezfooli for providing his research code for curvature analysis of decision boundaries of DCNs.

## Footnotes

[2]The output of the layer just before the softmax operation, commonly known as the logit layer.

[3]For more discussion about the implementation and associated concepts, refer to the supplementary material.

[4]The mean training-set orthogonal component $(\mathbf{I} - \mathbf{Q}_d \mathbf{Q}_d^\top)\bar{\mathbf{i}}$ can be added, but is approximately 0 in practice for data normalised by mean subtraction, as is the case here.

[5]In fact, this analysis is begun in [4], but only the singular *values* are examined.

[6]Measured as a percentage of samples from the dataset that undergo a change in their predicted label.

[7]Though not necessarily so for MNIST, because of its constraints: see supplementary material.

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
