[Supplementary Material]

# With Friends Like These, Who Needs Adversaries?

**Saumya Jetley**[*1]          **Nicholas A. Lord**[*1,2]          **Philip H.S. Torr**[1,2]

[1]Department of Engineering Science, University of Oxford
[2]Oxford Research Group, FiveAI Ltd.
`{sjetley, nicklord, phst}@robots.ox.ac.uk`

## A  Appendix

### A.1  Illustration of fundamental geometric argument

Figure 1: The left image illustrates the geometric argument of [1] connecting positive curvature to universal perturbations: from the perspective of a source image $\mathbf{i}$, if the actual class boundary (solid black) can be locally bounded on the outside by an arc (dashed black) of fixed positive curvature along a particular direction, then a perturbation along that direction large enough to cross the bounding arc *must* carry $\mathbf{i}$ across the actual boundary, changing its class. If such a direction is "shared" over many samples, it will accordingly represent a universal perturbation over that set. The right image illustrates that this effect is not particular to positive curvatures: in the negative-curvature case, the same holds true for an imagined point $\tilde{\mathbf{i}}$ on the other side of the boundary. The central image shows that a "flat" direction cannot have a material effect on membership of *either* class.

### A.2  Details and clarifications of use of differential geometric concepts

For a more involved treatment of the basic design of the decision boundary curvature analysis algorithm, consult the references [1] and [2] from which it is derived. If still in need of further clarification of fundamental concepts of differential geometry, consult a book such as [3]. We now address some specifics not discussed elsewhere, including in our own main text.

For one, it may seem strange to speak of the "curvature" of a decision boundary of a ReLU network, as analytically, this is everywhere either zero or undefined. "Curvature" is here (and in the relevant citations) computed via finite-difference numerical methods, implicitly corresponding to a smooth approximation of the actual piecewise linear boundary.

The astute reader may likewise notice that the use of the terms 'principal direction' and 'principal curvature' here (as originally in [1]) is somewhat relaxed. Let us clarify this point. Strictly speaking, the principal directions at a point on a manifold in an embedding space (such as an $(N-1)$-dimensional class decision boundary embedded in $N$-dimensional image space) are a local concept, forming an orthonormal basis spanning the space tangent to the manifold at that point. The principal curvature associated with each principal direction is the curvature, at the point of tangency, of the normal section of the manifold in the principal direction.

Generally, there is no reason to think that tangent spaces at different points on the boundary surface should coincide, and so *a priori*, it may not make any sense to speak of "principal directions" in the

---

[*]S. Jetley and N.A. Lord have contributed equally and assert joint first authorship.

embedding space. However, the authors of [1] are fully aware of this, and base their analysis on the hypothesis that there exist *image*-space directions which, when projected onto the respective tangent spaces of different points sampled from the boundary, correspond to similar curvature patterns. In other words, they assume that these directions are largely shared across sample images. A curvature can then be associated with each such direction by, for instance, taking the mean of the curvatures measured at those sample points. This is the relaxation of the terminology referred to above: the "principal directions" here represent a rotation of the canonical ($N$-dimensional) image-space axes, and the "principal curvature" associated with each direction represents the sample mean curvature measured along the tangent component of that vector at each sample point in a set.

In practice, the above can in fact be simplified, as is done explicitly in the version of the algorithm given in the main text as Alg. 1. Rather than managing the expense and complexity of the tangent-space projections, it suits our purposes here to work directly with the Hessian of the embedding function, effectively omitting the projection step.

Finally, we note that in contrast to the simplification given in the main text, the (very large) sample mean Hessians $\overline{\mathbf{H}}$ are never actually computed and stored explicitly. Instead, for each $\overline{\mathbf{H}}$, a function $f_{\overline{\mathbf{H}}}(\mathbf{v})$ is constructed that approximates $\overline{\mathbf{H}}\mathbf{v}$ via finite differences of backpropagated gradients. The MATLAB function *eigs* uses $f_{\overline{\mathbf{H}}}(\mathbf{v})$ to compute the eigendecomposition of the approximated $\overline{\mathbf{H}}$.

## A.3 Class identity as a function of perturbation scale: negative-curvature and flat directions

Figure 2: Selected class scores plotted as functions of the scaling factor $s$ of the perturbation along the most negatively curved directions and flat directions per net. The 'Median class score' plot compares the score of a randomly selected target class with the supremum of the scores for the non-target classes, for the negatively curved directions. For the flat curvature directions, it plots the score of a randomly selected target class and non-target class respectively. Each curve represents the median of the class scores over the associated dataset, bracketed below by the 30th-percentile score and above by the 70th. For the negative-curvature directions, the 'Transition out of target class' graph works in reverse to the corresponding positive-curvature graph in Fig. 2 of the main paper: 'sample proportion' represents the fraction of the dataset originally of the target class which retains the target-class label under perturbation, with the median softmax target-class score as before. The 'black dashed line' now represents the fraction of the dataset *not* originally of the target class which remains *outside* of the target class under perturbation. The images in the rightmost column illustrate a sample of these directions as visual patterns. Each block of eight images corresponds to the label (negative, or flat) to its left, and the two-image columns in each block correspond from left to right with the main four net-dataset pairs under study.

### A.4 Additional discussion, observations, and relationships with existing work

First, regarding the main result(s) of the paper as summarised in the conclusion, we would like to point out that the discovery of the universal adversarial perturbations of [4] strongly hinted at this outcome in advance. Those attacks are "universal" in precisely the sense that certain fixed directions perturb different images across class boundaries irrespective of the diversity in their individual appearances, *i.e.*, irrespective of the input components in all directions orthogonal to the perturbation. In fact, it is precisely *because* the functions are as separable as they are in this sense that the method used is able to identify them as well as it does. Note also that our results explain the "dominant label" phenomenon of [4], noted therein as curious but otherwise left unaddressed, in which a given universal perturbation overwhelmingly moves examples into a small number of target classes regardless as to original class identities. This is a manifestation of the targeting of particular classes by particular directions, a phenomenon so strong that it manifests in [4] *despite* the fact that their algorithm never explicitly optimises for this property.

In the context of discussing the adversarial vulnerability of DCNs, we would advise caution in using terms 'overfitting' and 'generalisation', particularly if done speculatively. Inspection of Fig. 3 (*top row*) and Fig. 4 (both in the main paper) will convince the reader that the directions of vulnerability produce fits that generalise very *well* to unseen data generated by the same distribution, observable as the near identity between training and test accuracy curves over the directions of highest curvature (or derivative) magnitude. This does *not* correspond to the classical notion of overfitting in machine learning. In fact, overfitting, observable as the divergence between the train and test error curves, happens over less salient (and thus, less vulnerable) directions. This can again be confirmed by inspection. The fact that the nets extract characterisations of their targets that do not correspond to that of humans is a separate issue.

Finally, the approximate symmetry of the feature response functions may appear counter-intuitive: why should the additive inverse of a given perturbation pattern produce such a similar result to the original pattern? We suggest that at least part of the explanation may rest in a fact long known in the hand-engineering of features for computer vision: natural image descriptors must typically neglect sign, because contrast inversion is a fact of the world. For instance, consider how a black bird and a white bird, which share the label *bird*, differ from one another when both are set against the background of a blue sky. One can revisit, for instance, [5] for a reminder. Empirically, this nonlinearity appears to be particularly important. Note that DCNs trained on MNIST are exceptional in that they may not adhere to this, as MNIST is unnatural in its fixing of contrast sign: *e.g.* the net need never learn that a black vertical stroke against a white background also represents the digit '1'.