[Reviews · NeurIPS 2018]

Reviewer 1



The work builds on the technique of [3] (detailed in Alg 1) to conduct a set of experiments that — demonstrate the sensitivity of networks along certain directions largely affecting its classification accuracy. In fact, projecting the image onto a small subspace consisting of these critical dimensions is sufficient to explain most of the model’s accuracy. The projection along these dimensions naturally turn out to be strong “features” for an image to be classified appropriately and therefore, techniques that find adversarial perturbations smartly exploit these dimensions. This paper makes the important observation that the very dimensions responsible for a good accuracy also cause vulnerability of the model to adversarial perturbations. I would like to know a more detailed response on previous work that claim to increase robustness to adversarial perturbations at negligible reduction in performance [29, 30] in this framework. What do the techniques proposed in these works correspond to in the framework studied in this paper ? Because — taking the view of this paper (and as mentioned in L238), it becomes infeasible to increase accuracy while increasing robustness to such attacks. The main negative of the paper is that it is hard to read due to 1) convoluted writing and 2) sloppy notation. Some pointers are below and in general, adding sufficient cross-references to previously defined terms will increase readability. Also, the graphs need to be presented more clearly — for eg. by moving the legend out of the plot. L16 — on one hand L156 — Algorithm 1, line 5 needs to be in two lines Section 3 — Since the rest of the paper critically depends on this section and Algorithm 1, this section must be presented more clearly. (like adequate comments in the algorithm box and grounding sentences in the section back to lines in Alg 1). Notation in the paper seems arbitrary. For example the set S_i is used to denote the subspace formed by the i-most positively curved directions and s is used to indicate samples from D in Alg. 1 (in the same section). Intuitively, one would assume s \in S and so, makes it a harder read. Fig 3 —Is there any intuition for why curves for CIFAR-100 differ from the rest (positive curvature) ?

Reviewer 2



The reason behind adversarial vulnerability of high dimensional classifier is still an open question. There are a few works trying to address this question by providing various and sometimes conflicting hypotheses. This work attempts to address the problem from a geometric perspective. The main hypothesis is that the “adversarial” directions are the ones which the classifier uses to distinguish between different classes. They provide some empirical evidence to support this idea. The paper raises some interesting points regarding adversarial robustness. It shows that the subspace spanned by the “curved” directions of the classifier in the image space captures the most relevant features for the classification task. Hence, projecting images on this subspace does not degrade the classification performance, while by projecting on the complement (flat subspace), the classifier accuracy drops significantly (Figure 4). Then, the authors argue that various adversarial perturbations are captured in the subspace of curved directions. Therefore, they conclude that adversarial directions are nothing but the discriminative features of different classes. This is indeed what happens in linear classifiers. I have to admit that I have not understood completely all the experiments. Honestly, I had hard time to follow some of the arguments as they were not rigorous enough. I feel that most of the arguments could be conveyed more clearly by equipping them with suitable math equations. Though this work has a lot of potential, it lacks the sufficient clarity and quality. For example, - The quality of plots are quite low, axes are missing, some numbers are even not rendered correctly. E.g., the numbers of the y-axes for CIFAR10 networks are missing. - Many definitions are missing and it is basically relied on the reader to guess them, e.g. N_f, N_p, N_n, etc. - Where does QR decomposition come from? I can guess but please be explicit. - What is Q_d? - What are the axes in Figure 1? x-axis? y-axis? To what does each image correspond? - How Alg 1 differs from that in [3]? why? - How do you compute the subspaces for ImageNet? One of the main drawbacks of the paper is that half of it is dedicated to prior works while it skips many important details regarding the contributions. Also, it is sometimes difficult to recognize their contribution from prior works. Though I really like the approach they take to study the adversarial vulnerability of deep classifiers, I guess the paper can heavily benefit from a complete rewriting. The most important change could be to shorten the prior works and to elaborate the contributions by taking the time to explain things using math equations. Anyway, the paper ends with the following sentence: “…nets’ favourite features are their worst adversaries”. This would be a great discovery if the authors could take a bit more time to polish their work.

Reviewer 3



The paper empirically studies phenomena associated with the principal curvature basis proposed in previous work. There are two main findings: 1) the classness scores are often monotonic on each side along the positive and negative curvature directions, while remain almost constant along the flat direction; 2) the positive and negative curvature directions characterize a subspace on which a neural network relies for classification. This further implies an interesting connection between classification power and sensitivity to adversarial perturbations. The paper presents interesting and novel observations. It reveals the contradiction between predictive performance and robustness. The experiments of subspace projection (Figures 4, 6, 7) have been designed in a smart way that substantiates the main claim of the paper. However, there are still many question remaining open. For example, it is not clear what causes such contradiction and whether it is a solvable problem. The authors have not addressed the "why" question even in the intuition level. As a result, although the findings might be interesting, it does not guide future research to devise more robust models. Ideally, I would love to see a better defense strategy coming out of the findings, which will make the paper much stronger. The presentation of the paper should be largely improved as it is very hard to follow. I would suggest adding formal definition of curvatures and other concepts, and also clearly describe each figure in detail, to make the paper more self contained. In the current version, it is very difficult to interpret the meanings of the plots.